# Genetic variation and RNA structure regulate microRNA biogenesis

Noemi Fernandez[1,*], Ross A. Cordiner[1,*], Robert S. Young[1], Nele Hug[1], Sara Macias[1,†] & Javier F. Cáceres[1]

MiRNA biogenesis is highly regulated at the post-transcriptional level; however, the role of sequence and secondary RNA structure in this process has not been extensively studied. A single G to A substitution present in the terminal loop of pri-mir-30c-1 in breast and gastric cancer patients had been previously described to result in increased levels of mature miRNA. Here, we report that this genetic variant directly affects Drosha-mediated processing of pri-mir-30c-1 *in vitro* and in cultured cells. Structural analysis of this variant revealed an altered RNA structure that facilitates the interaction with SRSF3, an SR protein family member that promotes pri-miRNA processing. Our results are compatible with a model whereby a genetic variant in pri-mir-30c-1 leads to a secondary RNA structure rearrangement that facilitates binding of SRSF3 resulting in increased levels of miR-30c. These data highlight that primary sequence determinants and RNA structure are key regulators of miRNA biogenesis.

---

[1] MRC Human Genetics Unit, Institute of Genetics and Molecular Medicine, Genome Regulation Section, Western General Hospital, University of Edinburgh, Edinburgh EH4 2XU, UK. * These authors contributed equally to this work. † Present address: Institute of Infection and Immunology Research, School of Biological Sciences, University of Edinburgh, King's Buildings, Edinburgh EH9 3FL, UK. Correspondence and requests for materials should be addressed to J.F.C. (email: Javier.Caceres@igmm.ed.ac.uk).

MicroRNAs (miRNAs) are short non-coding RNAs that negatively regulate the expression of a large proportion of cellular mRNAs, thus affecting a multitude of cellular and developmental pathways[1,2]. The canonical miRNA biogenesis pathway involves two sequential processing events catalysed by RNase III enzymes. In the nucleus, the microprocessor complex, comprising the RNase III enzyme Drosha, the double-stranded RNA-binding protein, DGCR8 and additional proteins carries out the first processing event, which results in the production of precursor miRNAs (pre-miRNAs)[3,4]. These are exported to the cytoplasm, where a second processing event carried out by another RNase III enzyme, Dicer, leads to the production of mature miRNAs that are loaded into the RISC complex[5].

Due to the central role of miRNAs in the control of gene expression, their levels must be tightly controlled. As such, dysregulation of miRNA expression has been shown to result in grossly aberrant gene expression and leads to human disease[6–9]. In particular, the microprocessor-mediated step of miRNA biogenesis has been shown to be regulated by multiple signalling pathways, such as the transforming growth factor-β pathway, leading to activation of subsets of individual miRNAs[10]. Furthermore, altered miRNA expression has been associated with the progression of cancer[11,12], where a global downregulation of miRNA expression is usually observed[13,14]. It was recently shown that miRNA biogenesis can also be regulated in a cell-density-dependent manner via the Hippo-signalling pathway, and that the observed perturbation of this pathway in tumours may

underlie the widespread downregulation of miRNAs in cancer[15]. Thus, miRNA production is tightly controlled at different levels during the biogenesis cascade. Extensive evidence has shown that RNA-binding proteins (RNA-BPs) recognize the terminal loop (TL) of miRNA precursors and influence either positively or negatively the processing steps carried out by Drosha in the nucleus and/or Dicer in the cytoplasm. These include the hnRNP proteins, hnRNP K and hnRNP A1, as well as the cold-shock domain protein, Lin28 and the RNA helicases, p68/p72 (ref. 16). In the case of the multifunctional RNA-BPs hnRNP A1, we have previously shown that it can act as an auxiliary factor by binding to the conserved TL of pri-miR-18a and promote its microprocessor-mediated processing[17,18]. Conversely, the same protein can act as a repressor of let-7 production in differentiated cells[19].

Several studies have shown that there is a correlation between the presence of polymorphisms in pri-miRNAs and the corresponding levels of mature miRNAs[20]; however, a mechanistic understanding of how sequence variation and RNA structure control miRNA biogenesis has not been explored in great detail. Screening of novel genetic variants in human pre-miRNAs linked to breast cancer identified two novel rare variants in the precursors of miR-30c and miR-17, resulting in conformational changes in the predicted secondary structures and leading to altered expression of the corresponding mature miRNAs. These patients were non-carriers of BRCA1 or BRCA2 mutations, suggesting the possibility that familial breast cancer

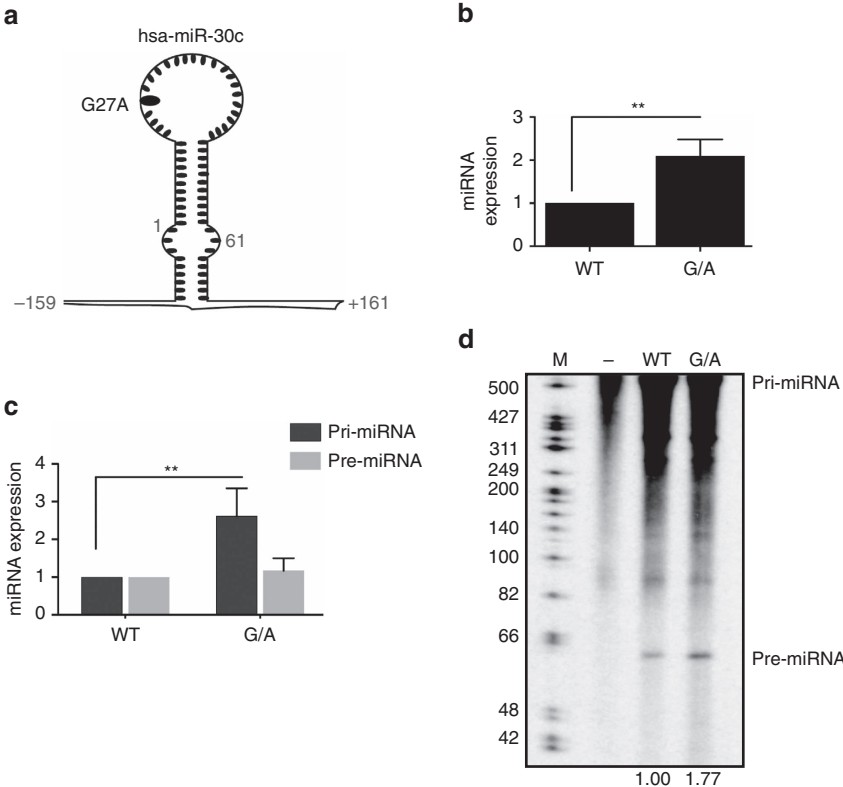

**Figure 1 | A genetic variant in the TL of hsa-pri-mir-30c-1 alters its normal expression. (a)** Schematic representation of the hsa-pri-mir-30c-1 transcript indicating the G to A mutation observed in breast and gastric cancer patients. Nucleotides (nt) encompassing primary (pri-miRNA; nt −159 to +161) and pre-miRNA (nt +1 to +61) used in this experiment are indicated. Numbers are relative to the first nt of the mature miRNA. **(b)** The relative expression levels of mature miR-30c in MCF7 cells transfected with plasmids encoding either a WT sequence or one bearing the G/A mutation (n = 12). **(c)** Levels of mature miR-30c in MCF7 cells transfected with in vitro-transcribed pri-mir-30c-1 (pri-miRNA) or an RNA oligonucleotide that mimics pre-miRr-30c-1 (pre-miRNA) sequences, either in a WT or G/A version (n = 5). Mann–Whitney U-test was used to evaluate differences between WT and G/A samples. Error bars indicate s.e.m. **P < 0.01. **(d)** Representative in vitro processing of pri-mir-30c-1 (380 nt) in MCF7 total cell extracts. Quantification of pre-miRNA band intensities are shown below and expressed as the relative intensity normalized to pre-miR-30c-1 WT variant.

may be caused by variation in these miRNAs[21]. In particular, the single-G to A substitution in primary miR-30c-1 (pri-mir-30c-1) TL, which was also later observed in gastric cancer patients[22], results in an increase in the abundance of the mature miRNA.

Here, we investigate the mechanism by which the pri-mir-30c-1 variant detected in breast and gastric cancer patients results in an increased expression of this miRNA. We found that this genetic variant directly affects the microprocessor-mediated processing of this miRNA. A combination of structural analysis with RNA chromatography coupled to mass spectrometry revealed changes in the pri-miRNA structure that lead to differential binding of a protein factor, SRSF3, that has been previously reported to act as a miRNA biogenesis factor. These results provide a mechanism by which the pri-mir-30c-1 genetic variant results in an increased expression of the mature miR-30c. Altogether these data highlight that primary sequence as well as RNA structure have a crucial role in the post-transcriptional regulation of miRNA biogenesis.

## Results

**Enhanced processing of pri-mir-30c-1 G/A variant.** To understand the mechanism underlying miR-30c deregulation in breast and gastric cancers, we investigated how the reported $G_{27}$-to-A mutation observed in a Chinese population might affect miRNA biogenesis. It was previously shown that this substitution results in an increase in the abundance of the mature miRNA; however, the mechanism that leads to an increased expression is unknown[21,22]. First, we transiently transfected MCF7 breast cancer cells with constructs encoding 380 nucleotides of primary hsa-pri-mir-30c-1 (pri-miRNA), either in a wild-type (WT) version or bearing the G/A variant (Fig. 1a). We observed that the G/A substitution resulted in increased levels of mature miR-30c (Fig. 1b), resembling the situation observed in patients with this mutation. Furthermore, this was not due to increased transcription of the G/A-harbouring pri-miRNA, as shown by unchanged pri-miRNA levels (Supplementary Fig. 1). To dissect the precise step of miRNA biogenesis pathway that is affected by the G/A substitution, we used an RNA version of pri-mir-30c-1

that has yet to undergo processing by the microprocessor in the nucleus and by Dicer in the cytoplasm (pri-miRNA). As a counterpart, we transfected an RNA oligonucleotide that mimics the pre-miRNA, a sequence that arises upon processing by the microprocessor. Importantly, we observed approximately a two-three fold increase of miR-30c mature levels when transfecting the G/A sequence derived from the pri-miRNA sequence, whereas no changes were detected following transfection of the pre-miRNA sequence (Fig. 1c). This experiment demonstrates that the G/A substitution exclusively affects the Drosha-mediated processing of the pri-miRNA. Moreover, we could recapitulate this result in an *in vitro* reaction. We found that *in vitro*-transcribed pri-mir-30c-1 was readily processed in the presence of MCF7 total extracts, rendering a product of ∼65 nucleotidess that corresponds to pre-mir-30c. Notably, the processing of the G/A variant was increased, when compared to the WT version, as was observed in living cells (Fig. 1d). The effect of the G/A variation in the processing of pri-mir-30c-1 was also recapitulated using a purified microprocessor (FLAG-Drosha/ FLAG-DGCR8 complex) and a shorter *in vitro*-transcribed substrate (153 nucleotides; Supplementary Fig. 2). Altogether, these complementary approaches indicate an enhanced microprocessor-mediated processing of the G/A variant sequence and this recapitulates what was previously observed in breast and gastric cancer patients.

**The sequence of pri-mir-30c-1 terminal loop is conserved.** Experiments described above confirmed a crucial role for the $G_{27}$ residue in influencing miR-30c biogenesis. We analysed the genomic variation in the hsa-pri-mir-30c-1 sequence across vertebrates, and detected substantial evolutionary constraint across the entire locus, as indicated by positive genomic evolutionary rate profiling (GERP) scores (Fig. 2a). Constrained residues, which highlight regions under purifying selection, are located in the mature miRNA sequences in both arms, as expected due to their effect in the regulation of gene expression. Interestingly, part of the TL, where $G_{27}$ is embedded, has also a very high level of constraint, which is suggestive of a role of this sequence in

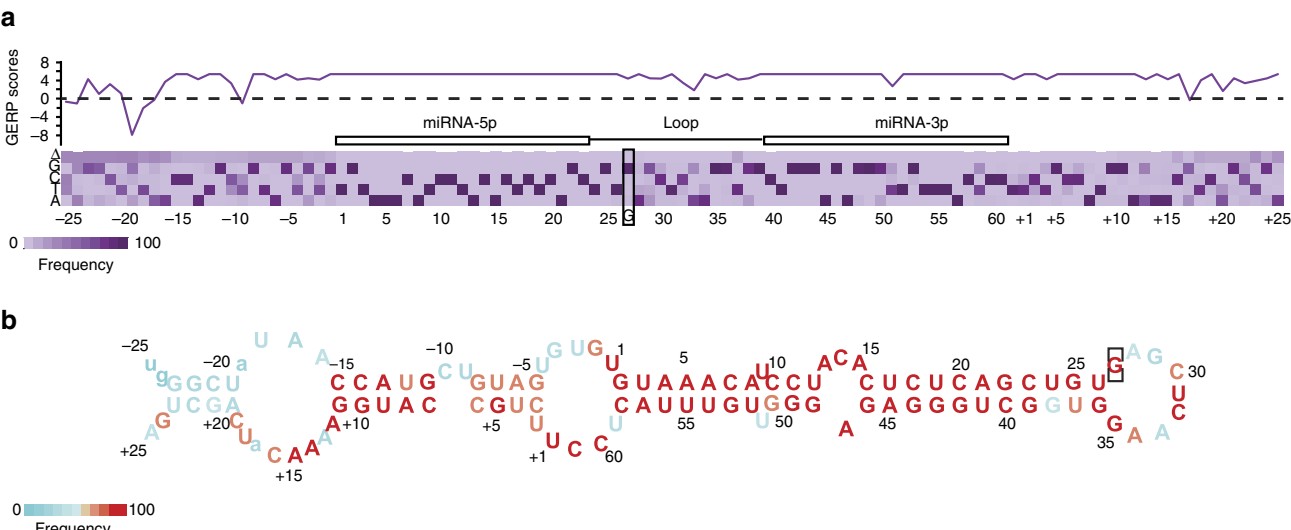

**Figure 2 | Sequence conservation of pri-mir-30c-1 precursors.** (**a**) Nucleotide-level GERP scores across the locus, indicating extensive evolutionary constraint. A GERP score above zero indicates significant constraint, while a score below zero indicates an excess of nucleotide substitutions beyond the expected neutral rate. The purple bars display the total number of observed nucleotide substitutions found in aligned sequences from 98 vertebrates. Δ represents absence of the nucleotide from 98 vertebrate sequences analyzed. The location of the modified G/A nucleotide is indicated by a rectangle. (**b**) Model of the predicted secondary structure with nucleotides coloured as in **a** to reflect their variation. Nucleotides present in <50% of the species are indicated in lower case.

miRNA biogenesis, as was previously described for a subset of miRNAs[16,18,23]. In addition, residues at the 5′end (nucleotides −1 to −8; −11 to −16 and −20 to −23) and the 3′end (nucleotides +1 to +16 and +18 to +25) are also highly constrained. Indeed, several of these residues were included as part of the stem in the *in silico* predicted RNA structure suggesting their importance for maintaining the RNA secondary structure (Fig. 2b). These data led us to focus our attention on these invariant sequences as potentially having a crucial role in the regulation of miR-30c biogenesis.

**Altered RNA structure of pri-mir-30c-1 G/A variant.** Next, to establish the importance of the $G_{27}$-to-A substitution in RNA structure, we performed structural analysis by selective 2′-hydroxyl acylation analysed by primer extension (SHAPE)[24]. This approach allows performing quantitative RNA structural analysis at single-nucleotide resolution and is mostly independent of base composition. While highly reactive residues are located at single- stranded regions, non-reactive nucleotides are involved in base pairs, non-Watson–Crick base pairs, tertiary interactions or single stacking interactions in the C2′-endo conformation[25]. To this end, *in vitro*-transcribed RNA comprising 380 nucleotides of pri-miR-30c-1 (either WT or the G/A variant sequence) was treated with *N*-methylisatoic anhydride (NMIA), which reacts with the 2′hydroxyl group of flexible nucleotides (Fig. 3a). Gross modifications of SHAPE reactivity were observed in specific regions of the G/A variant, when compared with the WT sequence. The resulting profiles revealed a decreased SHAPE reactivity in the TL (residues 28–30), with a concomitant increase in the 5′ region (−18, −16 and −15) as well as in the 3′end (nucleotides +11, +16 and +19; Fig. 3a,b). This result indicated the presence of different conformations in the pri-miRNA with the G to A substitution, as compared to its WT counterpart. To gain more information into the folding and tertiary structure of this pri-miRNA, we assessed the solvent accessibility of each nucleotide by hydroxyl radical cleavage footprinting, generated by reduction of hydrogen peroxide by iron (II)[26]. Hydroxyl radicals break the accessible backbone of RNA with no sequence dependence. We defined buried regions, as zones with more than two consecutives nucleotides having a reactivity ($R$) smaller than the mean of all reactivity, whereas exposed regions are those with more than two consecutive nucleotides having $R$ larger than the mean of all reactivity. We observed that the WT sequence presents two buried regions located between nucleotides +8 to +40 and +9 to +25, as well as two exposed segments between nucleotides −25 to +7 and nucleotides +40 to +8 (Supplementary Fig. 3a). The G to A substitution caused changes in the exposure to solvent, with both the TL (nucleotides +28 to +40) and also the 3′end region (nucleotides +17 to +22) becoming solvent accessible. By contrast, the 5′end (nucleotides −7 to +7) and a small region in miRNA-3p (nucleotides +55 to +58) are no longer solvent accessible.

Altogether, the SHAPE and radical hydroxyl data suggest that the $G_{27}A$ substitution is indeed affecting the RNA flexibility of pri-miRNA-30c-1, modifying both base-pairing interaction as well as solvent accessibility of the nucleotides located in the TL and in the basal region of the stem. This could be a consequence of a long-distance interaction disruption between those regions (Fig. 3c and Supplementary Fig. 3b), which could in turn modify the interaction with RNA-BPs important for miR-30c biogenesis.

**SRSF3 binds to a basal region of hsa-pri-mir-30c-1.** A working model is that either a repressor of microprocessor-mediated processing binds to the WT sequence or, alternatively, the change

in RNA structure induced by the G/A sequence variation could lead to the binding of an activator. To identify RNA-BPs that differentially bind to either the pri-miR-30c-1 WT or G/A sequence, we performed RNase-assisted chromatography followed by mass spectrometry in MCF7 total cell extracts[27]. This resulted in the identification of 12 proteins that interact with the WT sequence and 8 that bind to the G/A variant, being 7 common between both substrates. Significantly, several of the common proteins were previously implicated in miRNA biogenesis and/or regulation, including the heat shock cognate 70 protein[5], the hnRNP proteins, hnRNP A1 (refs 17,19) and hnRNP A2/B1 (ref. 28), the RNA helicase DDX17 (ref. 4), poly adenosine diphosphate ribose (ADP-ribose) polymerase 1 (PARP)[29], and the RNA-BPs fused in sarcoma/translocated in liposarcoma (FUS/TLS; Fig. 4a and Supplementary Fig. 4a). We could validate some of the interactors using immunoprecipitation followed by western blot analysis (Supplementary Fig. 4b).

FUS/TLS interacts with the WT sequence but it has been previously shown that it acts to promote miRNA biogenesis by facilitating the co-transcriptional recruitment of Drosha[30]. Thus, this would not be compatible with a repressive role of pri-mir-30c-1 WT processing. By contrast, in the case of the G/A variant, we identified a single exclusive interactor, SRSF3, which is a member of the SR family of splicing regulators. These family of proteins are involved in constitutive and alternative splicing, but some of them have been shown to fulfil other cellular functions[31,32]. Importantly, SRSF3 had been previously reported to be required for miRNA biogenesis[33], which would be compatible with a role as an activator of miRNA processing. Therefore, we focused on a putative role of SRSF3 as an activator of the processing of the G/A variant of pri-mir-30c-1. We could validate this interaction by RNA chromatography followed by western blot analysis with an antibody specific for SRSF3 (Fig. 4b). We also observed preferential binding of endogenous SRSF3 protein to the G/A variant of pri-miR-30c-1, as shown by immunoprecipitation of SRSF3 followed by quantitative reverse transcription PCR (RT-qPCR) quantification of the associated pri-miRNA (Fig. 4c). To analyse the interaction of SRSF3 with pri-miR-30c-1 WT and G/A sequences, we carried out toeprint and SHAPE assays using SRSF3 protein purified from MCF7 cells. Toeprint analysis was performed with fluorescent-labelled antisense primers and capillary electrophoresis[34]. In this assay, bound SRSF3 will block the reverse transcriptase and will illuminate the site where SRSF3 is bound to RNA. Prominent toeprint of SRSF3 with the G/A sequence was observed around nucleotides $A_{+25}$–$G_{+24}$ (RNA size 170 nucleotides, position $A_{+25}$; Fig. 4d). Similarly, analysis of SHAPE reactivity in the presence of added purified SRSF3, revealed a dose-dependent protection from NMIA attack upon addition of SRSF3 in specific RNA residues in a dose-dependent manner (nucleotides −19, −18, +16, +18, +19) of the basal region of G/A (Fig. 4f,e). Of importance, a conserved CNNC motif ((N signifies any nucleotide), nucleotides from +16 to +22), previously described as SRSF3 binding site[33,35,36] is located within the recognition place. Together, we can conclude that the interaction of SRSF3 with pri-miR-30c-1 takes place at the CNNC motif at the basal region of the G/A variant.

**SRSF3 increases processing of pri-mir-30c-1 G/A variant.** As previously described, SRSF3 was proposed to have a role in miRNA biogenesis by recognition of a CNNC motif located 17 nucleotides away from Drosha cleavage site[33]. Pri-mir-30c-1 has two overlapping CNNC motifs (residues from +16 to +21). Notably, accessibility around this region increased in the G/A variant, as determined by SHAPE and hydroxyl radical analysis

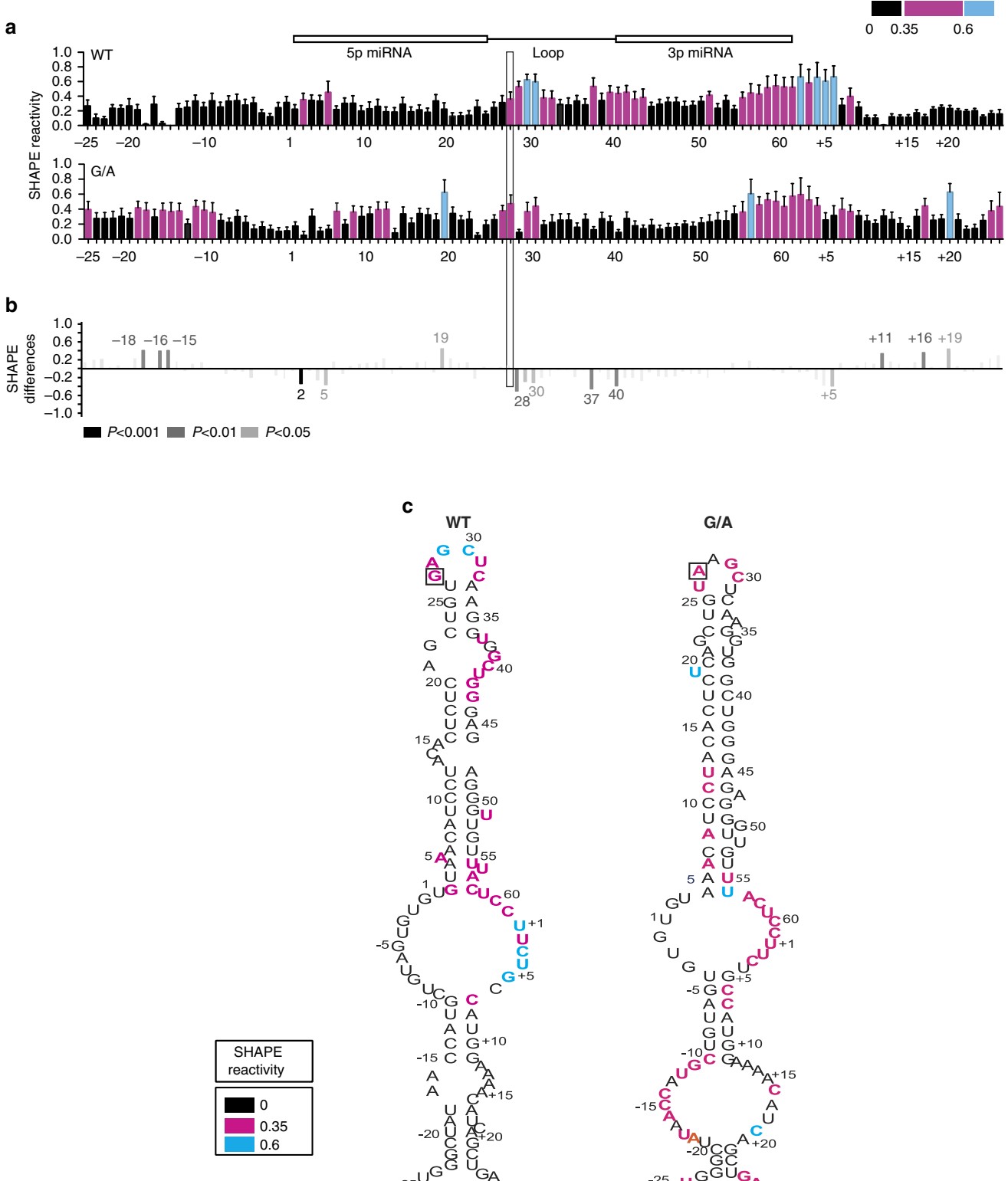

**Figure 3 | Structural analysis reveals the importance of the G to A substitution in pri-mir-30c-1.** (**a**) Values of SHAPE reactivity (depicted using colour coded bars) at each individual nucleotide position correspond to the mean SHAPE reactivity ( ± s.e.m.; $n = 10$). (**b**) SHAPE differences plots of the G/A variant relative to the WT sequence. Only nucleotides with absolutes differences bigger than 0.25 and statically significant are depicted (grey colours represent $P$ values). Bars above the basal line indicate increased SHAPE reactivity for the G/A variant, whereas those below the basal line indicate lower SHAPE reactivity. Mann–Whitney $U$-test was used to evaluate differences between WT and G/A samples. (**c**) Predicted secondary structures of WT and G/A variant sequences, estimated using RNAstructure software (see Methods), with nucleotides coloured to reflect their mean SHAPE reactivity values (**a**,**b**).

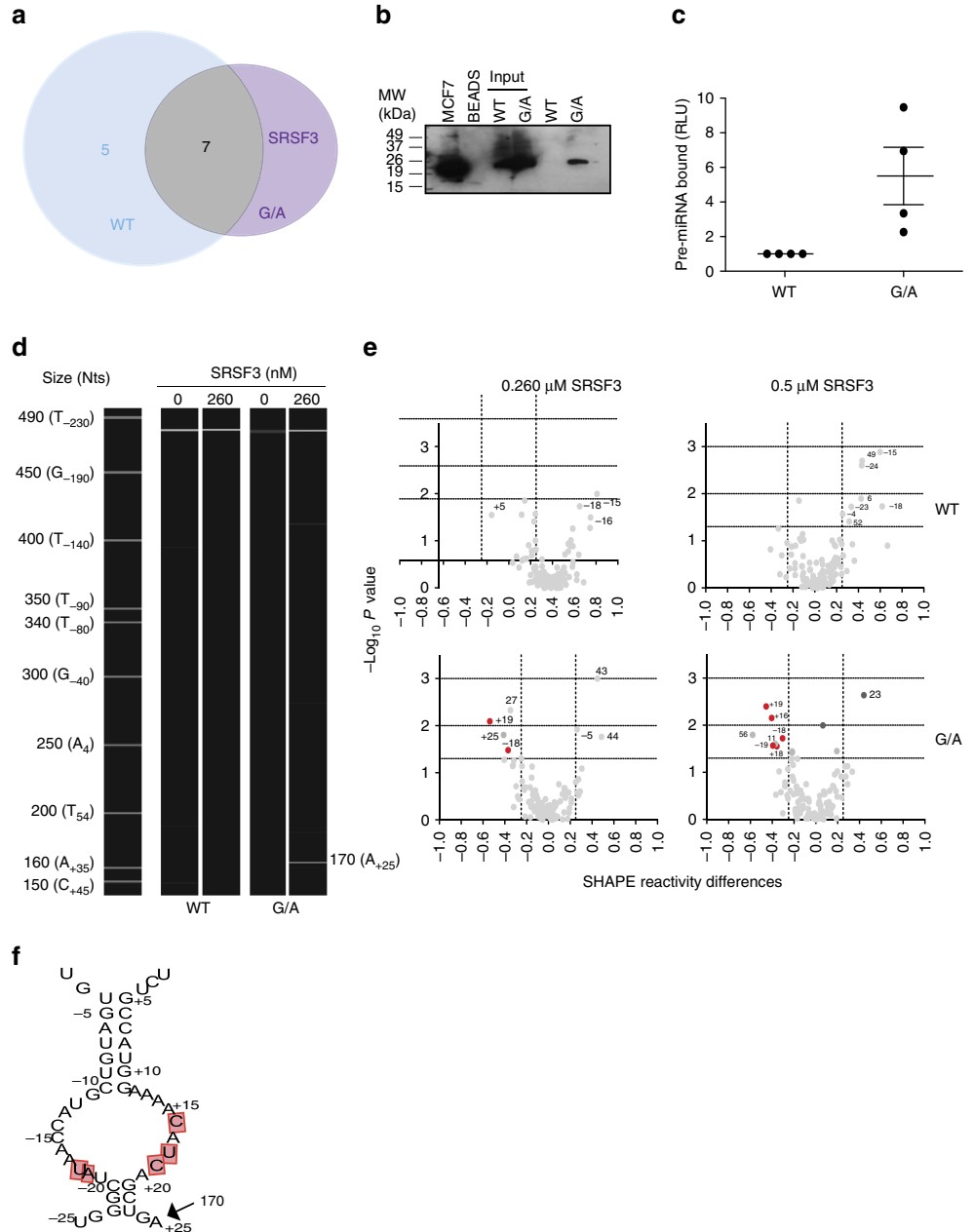

**Figure 4 | Identification of trans-acting factors binding to pri-mir-30c-1.** (**a**) Venn-diagram indicating the number of interactors observed for the WT sequence and the G/A variant. (**b**) Validation of the interaction of SRSF3 with the G/A variant sequence by RNA chromatography followed by western blot analysis with an antibody specific for SRSF3. (**c**) Immunoprecipitation with a specific anti-SRSF3 antibody followed by RT-qPCR quantification ( ± s.e.m.; $n = 4$). (**d**) Representative toeprinting assay in the presence or absence of 260 nM of purified SRSF3 protein using fluorescent-labelled primers and capillary electrophoresis. The unique RT stop in the G/A variant ($n = 2$) is indicated. (**e**) Volcano plots showing the SHAPE reactivity data in the presence of 260 nM (left panel) or 500 nM (right panel) of purified SRSF3 protein. The red dots indicate those positions protected by SRSF3 in a dose-dependent manner ($n = 5$). Mann–Whitney $U$-test was used to evaluate differences (**f**) Schematic representation of the basal stem of hsa-miR30c G/A illustrating the nucleotides protected by SRSF3 (in red) as well as the RT stop detected by toeprinting assay.

(Fig. 3 and Supplementary Fig. 3). Furthermore, toeprint and SHAPE assays in the presence of purified SRSF3 protein confirmed the specific recognition of the CNNC motif by SRSF3 in a dose-dependent manner (Fig. 4). Next, we addressed whether the preferential binding of SRSF3 to the pri-mir-30c G/A variant sequence was responsible for its increased expression by comparing the mature levels of miR-30c-1 WT or G/A variant under variable levels of SRSF3 expression. To this end, we co-transfected pri-mir-30c-1 constructs in MCF7 cells under transiently overexpress of SRSF3, or alternatively,

transfected specific siRNAs to knock-down endogenous SRSF3 protein (Supplementary Fig. 5a,b). Of interest, we observed that reduced levels of SRSF3 drastically decreased the levels of the G/A miR-30c variant, without affecting the levels of WT miR-30c (Fig. 5a, compare WT versus G/A panels). By contrast, transient overexpression of SRSF3 increased significantly the levels of WT mature miR-30c, but has a more modest effect on the G/A variant sequence. Altogether, these experiments suggest that SRSF3 binding is limiting in the WT scenario and that is essential to promote miRNA biogenesis in the G/A context.

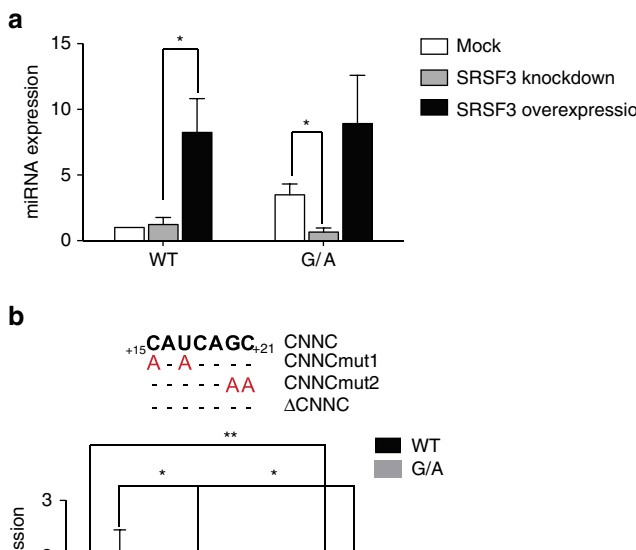

**Figure 5 | Role of SRSF3 in miR-30c expression. (a)** The relative expression levels of miR-30c WT and G/A in MCF7 cells with changing expression of SRSF3 protein. Error bars indicate s.e.m. ($n = 5$) *$P < 0.05$; **$P < 0.01$. **(b)** Mutational analysis of the CNNC motif. Nucleotides substitutions present in the CNNC region are shown in Red. Relative expression levels of miR-30c in MCF7 cells transfected with plasmids encoding either a WT sequence or one bearing the G/A mutation, and also including mutations in the CNNC motif, when indicated. Mann–Whitney U-test was used to evaluate differences. Error bars indicate s.e.m. ($n = 6$) *$P < 0.05$; **$P < 0.01$.

To confirm the role of SRSF3 in the differential processing observed with pri-mir-30c-1 G/A variant sequence, we proceeded to mutate the two consecutive CNNC motifs that are the natural binding sites for SRSF3 (Fig. 5b). We generated a set of mutants that affected either the first or second CNNC motif (mut1 and mut2, respectively) or a deletion of both motifs ($\Delta$CNNC). The CNNCmut1 carrying a double substitution $C_{+15}U_{+17}$ to AA, led to a severe reduction in the levels of miR-30c expression only with the G/A variant sequence (Fig. 5b). Similarly, a double substitution of the second CNNC motif $G_{+20}C_{+21}$ to AA (CNNCmut2) behaved similarly, exclusively affecting the G/A variant. Importantly, we were also able to show that a knockdown of SRSF3 expression affects the processing of pri-mir-30c-1 G/A variant, as expected; yet it does not compromise the processing of pri-mir-30c-1 G/A variant that lacks the CNNC motif (pri-mir-30c-1 G/A $\Delta$CNNC; Supplementary Fig. 5c). Thus, deletion of the CNNC motif in the G/A variant does indeed reduced processing of pri- to pre-miR-30c-1 G/A variant; however, the effect of decreasing SRSF3 is only seen when the CNNC motifs are present. Altogether, these experiments strongly suggest that the binding of SRSF3 is an important determinant of miR-30c expression. Finally, we could recapitulate the observation that SRSF3 binding is limiting for the processing of WT pri-mir-30c-1 in an *in vitro* system, supplemented with purified SRSF3 protein (Fig. 6). First, we found that the FLAG-Drosha/FLAG-DGCR8 complexes used for the *in vitro* processing assays contained residual levels of SRSF3 protein (Supplementary Fig. 6). Thus, the relative higher processing of the G/A variant can be

explained by the preferential binding of SRSF3 present in the reaction to this variant, as compared to the WT pri-mir-30c-1. Importantly, addition of purified SRSF3 protein increases the microprocessor-mediated production of WT pre-mir-30c-1 (Fig. 6a), whereas addition of purified SRSF3 to the G/A variant (Fig. 6c) or to $\Delta$CNNC variants that lack SRSF3 binding sites did not affect the processing activity (Fig. 6b,d). This is reminiscent of what was observed in MCF7 cells under variable levels of SRSF3.

## Discussion

The central role of miRNAs in the regulation of gene expression requires that their expression is tightly controlled. Indeed, the biogenesis of cancer-related miRNAs, including those with a role as oncogenes ('oncomiRs'), or those with tumour suppressor functions is often dysregulated in cancer[12,14,37]. Interestingly, some miRNAs have been shown to display both tumour suppressor and also oncogenic roles, depending on the cell type and the mRNA targets[38], as was described for miR-221, which exerts oncogenic properties in the liver[39] but also acts as a tumour suppressor in erythroblastic leukaemias[40]. Furthermore, miR-375 has been shown to display a dual role in prostate cancer progression, highlighting the importance of the cellular context on miRNA function[41].

Despite a more comprehensive knowledge on the role of RNA-BPs in the post-transcriptional regulation of miRNA production, there is only circumstantial evidence on how RNA sequence variation and RNA structure impact on miRNA processing. There are several reports showing that a single-nucleotide substitution in the sequence of pre-miRNAs could have a profound effect in their biogenesis. Nonetheless, there is limited information about single-nucleotide polymorphisms (SNPs) in the TL region of pri-miRNAs. A bioinformatic approach led to the identification of 32 such SNPs in 21 miRNA loop regions of human miRNAs[42]. Some studies have found a correlation between the presence of polymorphisms in pri-miRNAs and expression levels of their corresponding mature miRNAs, affecting cancer susceptibility, as shown for miR-15/16 in chronic lymphocytic leukaemia[43,44], miR-146 in papillary thyroid carcinoma[45] and miR196a2 in lung cancer[46]. Another example is the finding of a rare SNP in pre-miR-34a, which is associated with increased levels of mature miR-34a. This could be of biological significance since precise control of miR-34a expression is needed to maintain correct beta-cell function, thus this could affect type 2 diabetes susceptibility[47]. The emerging picture is that human genetic variation could indeed not only have a role in miRNA function by affecting miRNA seed sequences and/or miRNA binding sites in the 3'UTRs of target genes, but it can also contribute significantly to modulation of miRNA biogenesis[20].

In this study, we focused on a rare genetic variation found in the conserved TL of pri-mir-30c-1 ($G_{27}$-to-A variant) that was found in breast cancer and gastric cancer patients and leading to increased expression of miR-30c (refs 21,22). There is circumstantial evidence that miR-30c is involved in many malignancies acting as tumour suppressor[48,49] or as an oncogene[50–52]. Indeed, predicted targets of miR-30c, include phosphate and tensin homologue (PTEN) and ataxia telangiectasia mutated (ATM) that represent key breast cancer tumour suppressor genes[21]. Interestingly, a significantly increased risk of gastric cancer was observed in subjects with the homozygote AA of pre-miR-30c, when compared with heterozygote AG or homozygote GG carriers[22]. To understand the mechanism underlying miR-30c deregulation in breast cancer, we investigated how this mutation affects miRNA biogenesis. We show that the G–A substitution in pri-mir-30c-1 directly affects Drosha-mediated processing both *in vitro* as well as in cultured

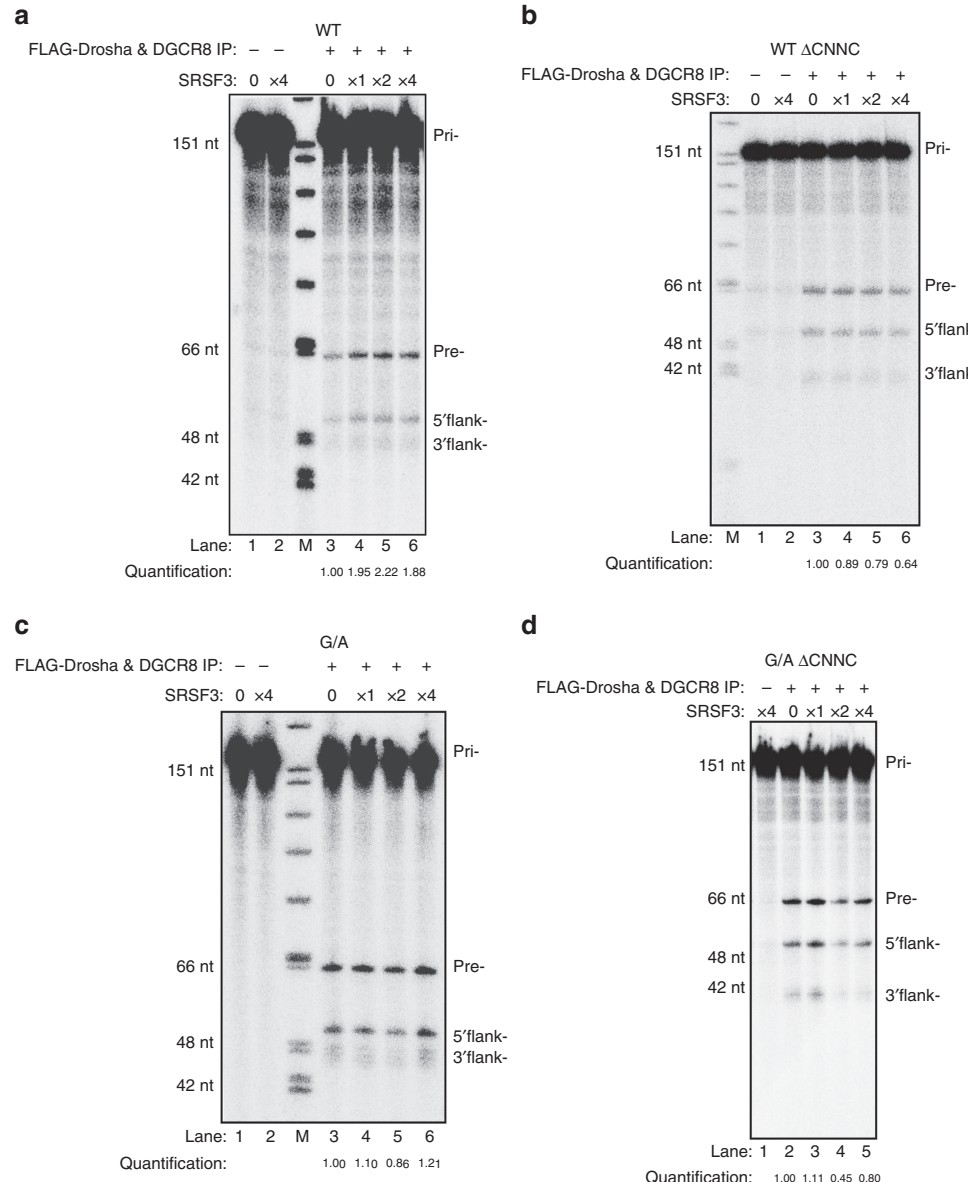

**Figure 6 | Representative *in vitro* processing assays of pri-miR-30c-1 variants. (a,b)** Processing of radio-labelled shorter WT pri-mir-30c-1 comprising or lacking SRSF3 binding sites (WT and WT ΔCNNC, respectively) in the presence of FLAG-tagged complexes immunopurified from HEK293T cells transiently co-expressing FLAG-Drosha/FLAG-DGCR8 ( + ) or FLAG-empty vector control ( − ) with addition of purified SRSF3 protein (1 × corresponds to 65 nM of protein). **(c,d)** Processing of pri-mir-30c-1 G/A variant comprising or lacking SRSF3 binding sites (G/A and G/A ΔCNNC, respectively), as described above. Quantification of pre-miRNA band intensities are shown below and expressed as the relative intensity normalized to *in vitro* processing assay in the absence of purified SRSF3 protein (lane 3 in each case).

cells (Figs 1,5,6 and Supplementary Fig. 2). The conservation of pri-mir-30c-1 sequences across vertebrate species highlights the importance of the primary sequence in the TL, 5′ and 3′ regions (Fig. 2), suggesting a crucial role in miRNA biogenesis. Indeed, conserved sequences in TL have been shown to be important for recognition by auxiliary factors[23] as well as for DGCR8 binding[53], allowing efficient and accurate miRNA processing. It has also been shown that pri-miRNA tertiary structure is a major player in the regulation of miRNA biogenesis, as observed for the well characterized miR17-92 cluster[54–56]. Here, using SHAPE structural analysis, in conjunction with solvent accessibility analysis by hydroxyl cleavage, we found that the G/A sequence variation leads to a structural rearrangement of the apical region of the pri-miRNA affecting the conserved residues placed at the basal part of the stem (Fig. 3 and Supplementary Fig. 3). This

demonstrates that pri-mir-30c-1 is organized as a complex and flexible structure, with the TL and the basal region of the stem potentially involved in a tertiary interaction. Further work is required to determine the existence of direct contacts between these regions.

Interestingly, we also observed that this RNA structure reorganization promotes the interaction with SRSF3, an SR protein family member that was demonstrated to facilitate pri-miRNA recognition and processing[33], by recognizing the CNNC motif located 17 nucleotides away from Drosha cleavage site. A recent study aiming to identify novel determinants of mammalian primary microRNA processing confirmed that the CNNC primary sequence motif selectively enhances the processing of optimal-length hairpins. This study, also predicted that a fraction of human SNPs will lead to alterations of pri-miRNA processing[57].

Pri-mir-30c-1 has two overlapping CNNC motifs (residues from $+16$ to $+21$; Fig. 5b). Notably, accessibility around this region increased in G/A variant (Fig. 3). Furthermore, toeprint and SHAPE assays in the presence of purified SRSF3 protein clearly demonstrated that SRSF3 is specifically recognizing the CNNC motif in a dose-dependent manner (Fig. 4). Altogether, data presented here suggest that binding of SRSF3 to the WT sequence is limiting and that the structural reorganization induced by the G/A substitution makes the SRSF3 binding sites more accessible. Taking everything into account we propose a model whereby a genetic variant in a conserved region within the TL of pri-mir-30c-1 causes a reorganization of the RNA secondary structure promoting the interaction with SRSF3, which in turn enhances the microprocessor-mediated processing of pri-mir-30c-1 leading to increased levels of miR-30c (Fig. 7). We conclude that primary sequence determinants and RNA structure are key regulators of miRNA biogenesis.

## Methods

**Plasmids constructions.** A pri-mir-30c-1 construct was amplified from human genomic DNA by PCR with specific primers 30c1s (5′-CAAGTGGTTCTGTG TTTTTATTG-3′) and 30c1a (5′-GTACTTAGCCACAGAAGCGCA-3′) The PCR product was digested with *EcoR*I and was subsequently cloned into the pCDNA3.1 vector (ThermoFisher). The $G_{27}$–to-A mutation was generated by a two-step PCR strategy. First, pri-miR-30c-1 was amplified with a 30Cmut1 oligo (5′-CCTTGAGCTTACAGCTGAGAG-3′) and 30c1s and with 30Cmut2 oligo (5′-CTCTCAGCTGTAAGCTCAAGG-3′) and 30c1a. Both PCR products, were purified (Qiagen), pooled and used as a template for amplification with 30c1s and 30c1a primers. The resulting PCR product was cloned in pGEMt (Promega). pGEMt G/A plasmid was digested with *EcoR*I for cloning into pCDNA3.1. The CNNC motifs were subjected to site-specific mutagenesis by PCR amplification[58]. Briefly, 10 ng of pri-miR30c-1 (WT or G/A) was PCR amplified with the desired mutagenic oligonucleotide (mut1: 5′-CTTCATTTGATGTTTTCCATGGC-3′, mut2: 5′-CTTCTTTTTTTTTTTTCCATGGC-3′ or CNNC 5′-CTTCAGATG TTTTCCATGGC-3′) and CNNCs primer (5′-CTGCTTACTGGCTTATCG-3′). The PCR product was cleaned (Qiagen) and used as the 5′-flanking primer in a second PCR with an equal molar amount of primer CNNCa (5′-GATATCT GCAGAATTCACTAG-3′).

The product of the second PCR was digested with *EcoR*I (New England Biolabs), purified by agarose gel electrophoresis and ligated to the large *EcoR*I fragments of pri-mir-30c-1 to produce the desired constructs (CNNCmut1, mut2 and ΔCNNC). All the sequences were confirmed by automatic sequencing. A list of oligonucleotide sequences used in this study is presented as Supplementary Table 1.

**Cell culture.** MCF7 and HEK 293 T cells were grown in high glucose Dulbecco's modified Eagle's medium (Invitrogen) supplemented with 10% (v/v) fetal calf serum (Invitrogen) and penicillin-streptomycin (Invitrogen) and incubated at 37 °C in the presence of 5% $CO_2$. Cells were tested for mycoplasma contamination.

**Transfections.** MCF7 cells grown in 24-well plates were transfected with either pri-mir-30c-1 (WT or G/A) constructs, *in vitro*-transcribed RNA (0.33 μg per $10^5$ cells) or oligonucleotides-encoding pre-mir-30c (Sigma Aldrich)(5′-UGUAAAC AUCCUACACUCUCAGCUGU**G**AGCUCAAGGUGGCUGGGGAGAGGGUUG UUUACUCC-3′) using Attractene (Qiagen), following manufacturer's instruction. pCDNA-3, pri-miR-30a and oligo30a (5′-UGUAAACAUCCUCGACUGGAAGC UGUGAAGCCACAAAUGGGCUUUCAGUCGGAUGUUUGCAGC-3′) were used as negative controls in DNA or RNA transfections, respectively. Cells extracts were prepared at 8 or 48 h (after RNA/DNA addition) by direct lysis using 100 μl of lysis buffer (50 mM Tris-HCl at pH 7.8, 120 mM NaCl, 0.5% NP40). For SRSF3 gene silencing/overexpression MCF7 cells grown in 15 cm dishes were transfected with ON-TARGETplus siRNA (Dharmacon) or pCG. As negative controls, cells were transfected with ON-TARGETplus Non-targeting siRNAs and pCG plasmid, respectively. Cells were split in 24-well dishes 24 h after transfection and 24 h later transfected with different versions of pri-miRNA constructs. HEK 293Ts were grown to 70% confluency in 6-well plates and then transiently co-transfected with 3 μg of FLAG-Drosha and 1 μg of FLAG-DGCR8, or 4 μg FLAG-empty vector per well. Cells were expanded for 24 h, then split to 10 cm plates and expanded for a further 24 h before cells were scraped, collected and snap frozen until required.

**Western blot analysis.** Equal amounts of total protein (determined by Bradford assay) were loaded in 12% NUPAGE gels (Invitrogen) and transferred to cellulose membranes using IBLOT system (Invitrogen). Identification of SRSF3 was performed with a rabbit polyclonal antibody (RN080PW, Medical and Biological Laboratory, (MBL), Dilution 1:500), followed by a secondary horseradish peroxidase-conjugated antibody and ECL detection (Pierce). Other primary antibodies used in this study were: mouse monoclonal anti-PARP-1 antibody (E-8): sc-74469, Santa Cruz Biotechnology, Dilution 1:500 ); Rabbit polyclonal anti-DDX17 antibody ((S-17): sc-86409, Santa Cruz Biotechnology, Dilution 1:500); Rabbit polyclonal anti-hnRNP A1 antibody (PA5-19431, Invitrogen antibodies, Dilution 1:500); Rabbit polyclonal anti-TLS/FUS antibody (ab23439, Abcam, Dilution 1:1,000).

***In vitro* transcription of pri-miRNA substrates.** Before RNA synthesis, pri-mir-30c-1 plasmids (WT and G/A variant) were linearized using *Apa*I (New England Biolabs). In addition, shorter pri-mir-30c-1 probes were PCR amplified from pri-mir-30c-1 plasmids (WT, G/A and their respective ΔCNNC counterparts) for *in vitro* processing assays (Fig. 6 and Supplementary Fig. 1) using a forward primer harbouring a T7-promoter sequence fused to a 19 nucleotide sequence complementary to pri-mir-30c-1 (5′-AATACGACTCACTATAGGGCTGATCAACC CTGGACC-3′) and a reverse primer (5′- AGTGGAGACTGTTCCTTCTTC-3′), which when *in vitro*-transcribed generated a 153 nucleotides (CNNC) or 146 nucleotides (ΔCNNC) and then subsequently PCR purified (Qiagen) for *in vitro* transcription. An aliquot of 1 μg of DNA or 400 ng of PCR product were *in vitro* transcribed for 1–2 h at 37 °C using 50 U of T7 RNA polymerase (Roche) in the presence of 0.5 mM ribonucleoside tri-phosphates (rNTPs) and 20 U of RNAsin (Promega). When needed, RNA transcripts were labelled using (α-$^{32}$P)-UTP (800 Ci per mmol, Perkin Elmer) and following DNASe treatment, unincorporated $^{32}$P-UTP was eliminated by exclusion chromatography in TE equilibrated columns (GE Healthcare) or PAGE gel purification, followed by phenol/chloroform and ethanol precipitation.

**SRSF3 purification.** Purification of SRSF3 from MCF7 cells was performed following transient expression of epitope-tagged SRSF3 (ref. 59). Briefly, MCF7 cells were grown to confluence in 15 cm dishes and transiently transfected with pCG-T7-SRSF3 (ref. 60). Forty-eight hours after transfection cell pellets were resuspended in 20 ml of ice-cold lysis buffer (50 mM NaP buffer, pH8, 0.5 M Na Cl, 5 mM b-glycerophosphate, 5 mM KF, 0.1% Tween 20 and 1 × protease inhibitor cocktail) and sonicated five times 30 s followed by centrifugation at 16,000 *g* for 20 min at 4 °C. After centrifugation the supernatant was loaded into a chromatography column (Biorad) previously prepared with T7 Tag antibody Agarose (Novagen). The flow-through was collected and loaded a second time. The column was washed two times with 10 ml of lysis buffer. Then, eluates were eluted with 10 serial 0.8 ml volumes of elution buffer (0.1 M citric acid, pH 2.2, 5 μm β-glycerophosphate c and 5 mM KF) and collected in microcentrifuges tubes containing 200 μl of 1 M Tris pH 8.8 mixed and stored at 4 °C. The fractions were

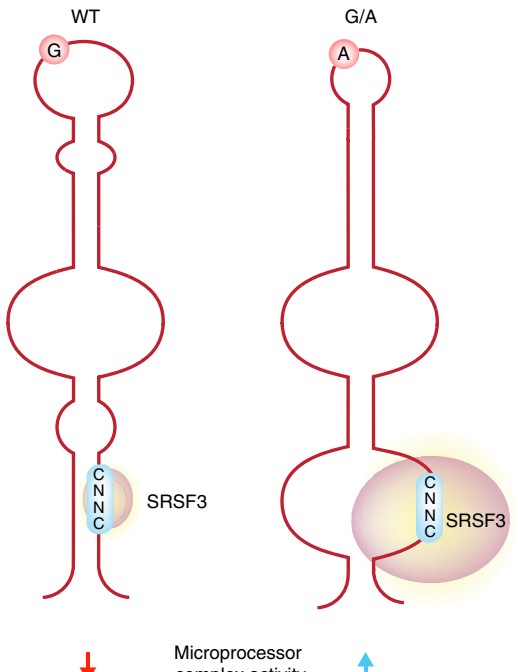

WT        G/A

Microprocessor
complex activity

**Figure 7 | Model depicting the stimulatory effect of the G/A genetic variant on pri-miRNA processing.** Cartoon depicting a model whereby the genetic variant identified in pri-mir-30c-1 leads to a secondary RNA structure rearrangement that facilitates binding of SRSF3 resulting in increased microprocessor-mediated processing of pri-mir-30c-1.

analysed on SDS–PAGE (Invitrogen) followed by Coomassie blue staining. The eluates containing the protein were dialyzed overnight against BC100 buffer (20 mM tris, pH8, 100 Mm KCl, 0.2 mM EDTA pH8 and 20% glycerol), and stored in aliquots at − 80 °C.

**In vitro processing assays.** Radio-labelled *in vitro*-transcribed pri-mir-30c-1 (50,000 c.p.m.) was incubated with 650 µg of MCF7 total cell extract[61] (Fig. 1d), or incubated with FLAG-tagged complexes immunopurified from HEK293T (ATCC CRL-3216) cells transiently co-expressing FLAG-Drosha & FLAG-DGCR8 or FLAG-empty vector control, in the absence or RNase[62] (Fig. 6 and Supplementary Fig. 2). Additionally, *in vitro* processing reactions were supplemented with increasing concentrations of immunopurified T7-SRSF3 (Fig. 6). The *in vitro* processing reactions were performed in the presence of buffer A (0.5 mM adenosine triphosphate (ATP), 20 mM creatine phosphate and 6.4 mM MgCl$_2$). Reactions were incubated for 1 h at 37 °C and treated with proteinase K. RNA was extracted by phenol/chloroform and ethanol precipitation. Samples were resolved in an 8% 1 × TBE polyacrylamide urea gel. An uncropped scan of the experiment corresponding to Fig. 1d is provided as Supplementary Fig. 7a.

**Pre-miRNA and mature miRNA qRT-PCR.** One step qRT-PCR was used to calculate pre-miR-30c-1 levels. Specifically, 300 ng of total RNA was used with SuperScript III Platinum SYBR Green One-Step qRT-PCR Kit (Invitrogen, 11736-051) on CFX96 real time system. Primers located within the precursor sequence of miR30c (Fwd 5′-TGTAAACATCCTACACTCTCAG-3′; Rev – 5′-GAGTAAAC AACCCTCTCCCA-3′) or within the primary sequence of miR30c (fwd 5′-CA GTGGTCAGGGGCTGAT-3′; rev 5′-GGAGTGGAGACTGTTCCTTCT-3′) were used to calculate pre-miR30c levels[63]. The relative amounts of mature miR-30c present in total RNA samples were measured using Exiqon miRCURY LNA following manufacturer's instructions. Total RNA from cytoplasmic lysates was isolated using RNAzol (Invitrogen) and RT was carried out with Universal RT microRNA PCR (Exiqon) using 500 ng of RNA. A 1/20 dilution of the RT reaction was used for qPCR with a specific microRNA LNA (Exiqon) primers set and the LightCycler system with the FastStart DNA Master Green II (Roche). The amount of miR-30c detected in the reaction was normalized by a parallel reaction performed with U6 and SNOR48 primers. For SRSF3 gene silencing or overexpression, MCF7 cells (ATCC HTB-22) grown in 15 cm dishes were transfected with ON-TARGET plus siRNA (Dharmacon) or pCG. As negative controls, cells were transfected with ON-TARGETplus Non-targeting siRNAs and pCG-T7 plasmid, respectively. Cells were split in 24-well dishes 24 h after transfection and 24 h later transfected with different versions of pri-miRNA constructs.

**Quantitative RNA co-immunoprecipitation.** Lysates from pri-miR-30c-1 (WT and G/A) transfected cells were pre-cleared with mouse IgG beads followed by incubation with a polyclonal rabbit anti-SRSF3 antibody (MBL). The complexes were pulled-down using protein G beads (Amershan), then treated with proteinase K (Sigma) and RNA was extracted and purified using Trizol (Invitrogen)[64]. qRT-PCR was carried out with superscriptIII (Invitrogen) RT using 500ng of RNA as a template and the TL (5′-GGAGTAAACAACCCTCTCCCAGC-3′) and actin (5′-GGTCTCAAACATGATCTGGG-3′) primers. Then a 1/20 dilution of the RT reaction was used for qPCR with appropriate pairs of specifics primers (TL: 5′-CA TCCTACACTCTC-3′ and 5′-GGAGTAAACAACCCTCTCCCAGC-3′, actin: 5′-GGGTCAGAAGGATTCCTATG-3′ and 5′-GGTCTCAAACATGATC TGGG-3′). Quantitative RNA co-immunoprecipitation values were calculated according to the formula: qPCR value TL mRNA immunoprecipitation/qPCR value TL mRNA total)/(qPCR value control mRNA immunoprecipitation/qPCR value control mRNA total). An uncropped scan of the experiment corresponding to Fig. 4b is provided as Supplementary Fig. 7b.

**Phylogenetic conservation.** The alignment of 98 vertebrate sequences of pri-mir-30c-1 were retrieved from USCS genome browser. Each nucleotide frequency was plotted in a heat map.

**Evolutionary constraint.** Evolutionary constraint was quantified for individual nucleotide positions as the number of rejected substitutions, as calculated by the GERP + + algorithm. This data was extracted from the UCSC genome browser, where they had been calculated over their 36-way mammalian genome alignments.

**RNA structure analysis.** RNA structure models as shown on Figs 2b and 3c were obtained using RNAstructure software (http://rna.urmc.rochester.edu/RNAstructureWeb/).

**SHAPE analysis.** Pri-mir-30c-1 RNA was treated with NMIA (Invitrogen), as the modifying agent[65]. For primer extension, equal amounts of NMIA treated and untreated RNA (0.5 pmols) were incubated with 2 pmol of 5′-end fluorescently labelled primer (5′-CTAGATGCATGCTCGAGCG-3′). NED fluorophore was used for both, treated and not treated RNAs while VIC fluorophore was used for the

sequencing ladder. cDNA products were resolved by capillary electrophoresis. Pri-miR-30c-1-SRSF3 complexes were assembled in folding buffer (100 mM HEPES pH 8, 6 mM MgCl$_2$) using 170 nM RNA in the presence of increasing amounts of purified SRSF3 protein (260 and 500 nM). Then, RNA alone or pre-incubated with SRSF3 was treated with NMIA. RNA was phenol extracted and ethanol precipitated and then subjected to primer extension analysis.

**Hydroxyl radical footprinting.** Pri-mir-30c-1 RNA was subjected to Hydroxyl radical footprinting. Briefly, 1.7 pmol of RNA was denatured and folded in folding buffer (40 mM MOPS pH 8.0, 80 mM KOAc, and 0.5 or 6 mM MgCl$_2$). Samples were incubated with 1 µl of the Fe(II)–EDTA complex, 1 µl of sodium ascorbate and 1 µl of hydrogen peroxide for 30 s at 37 °C. Fe(II)–EDTA (7.5 mM Fe(SO$_4$)2(NH$_4$)2 · 6H$_2$O and 11.25 mM EDTA, pH 8.0), 0.3% hydrogen peroxide and 150 mM sodium ascorbate solutions were freshly prepared. As a control a lacking Fe(II)–EDTA reaction was performed[66]. Samples were quenched and precipitated by addition of one-third of 75% glycerol, 1 µl of 20 mg ml − 1 glycogen, 1 µl of 3 M NaCl, 2 µl of 0.5 M EDTA and 2.5 volumes of ice-cold ethanol. RNAs were re-suspended and reverse-transcribed using fluorescent primers as described for SHAPE reactivity. cDNA products were resolved by capillary electrophoresis.

**SHAPE reactivity and Hydroxyl radical cleavage data analysis.** SHAPE electropherograms of each RT were analysed using the quSHAPE software[67]. Then, data from 10 independent assays were used to calculate the mean of SHAPE reactivity. To minimize the technical variation, each RT stop quantitative value was normalized to the total intensity of the average of 10 highly reactive nucleotides in the reaction. Grubbs' test was used to identify outliers. To obtain SHAPE differences between pri-mir-30c WT and GA or Free RNA and RNA-SRSF3 complexes, the SHAPE reactivity values obtained in WT or free RNA were subtracted from the reactivity values obtained in GA or in the presence of the protein. The statistical analysis was performed by Mann–Whitney U-test. Only nucleotides with absolutes differences larger than 0.25 and statistically significant were considered. Hydroxyl radical cleavage intensities of each reaction were also analysed using quSHAPE software[66]. Then, data from six independent assays were used to calculate the mean of hsa-pri-mR-30c WT or GA cleavage. Grubbs' test was used to identify outliers. Buried regions were defined as zones with more than two consecutives nucleotides having a reactivity ($R$) smaller than the mean of all reactivity, whereas exposed regions are those with more than two consecutive nucleotides having $R$ larger than the mean of all reactivity.

**Toeprint assays.** Pri-mir-30c-1: SRSF3 complexes were assembled as described for SHAPE analysis. After protein incubation samples were subsequently subject to primer extension using fluorescent primer (NED) 5′-CTAGATGCATGCTCGAGCG-3′.pri-miR-30c (ref. 34). Primer extension products were extracted with phenol, and ethanol precipitated pellets were resuspended in 5 µl of HI-Di formamide (ThermoFisher), which included 0.5 µl of GeneScan 500 Liz dye size standards. The products were separated by capillary electrophoresis and analysed by GeneMarker software.

**RNA chromatography.** RNase-assisted RNA chromatography with RNAse A/T1 was performed, using *in vitro*-transcribed pri-mir-30c-1 and total MCF7 cell extracts[28]. RNA-bound proteins were separated using 4–12% NUPAGE bis–tris system (Invitrogen) and individual lanes were subjected to mass spectroscopy (BSRC Mass Spectrometry and Proteomics Facility, University of St Andrews). Results were confirmed by western blot analysis with anti-SRSF3 (MBL), anti-PARP-1 (Santa Cruz), anti-DDX17 (Santa Cruz), anti-hnRNPA1 (Thermo Scientific) and anti-FUS (Abcam) specific antibodies, as indicated above (Western blot analysis section).

**Data availability.** The data that support the findings of this study are available from the corresponding author upon reasonable request.

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

## Acknowledgements

We are grateful to Magdalena Maslon (MRC HGU) for discussions and to Javier Martinez (IMBA, Vienna) and Encarnación Martínez-Salas (CBM, Madrid) for critical reading of the manuscript. This work was supported by Core funding from the Medical Research Council and by the Wellcome Trust (Grant 095518/Z/11/Z).

## Author contributions

N.F., R.A.C., S.M. and J.F.C. conceived, designed, interpreted the experiments and wrote the manuscript. N.F. performed biochemical and structural experiments, R.A.C. carried out *in vitro* pri-miRNA processing and N.H. performed biochemical experiments. S.M. carried out some of the initial experiments. R.S.Y. provided bioinformatics analysis. J.F.C. supervised the whole project.

## Additional information

**Competing interests:** The authors declare no competing financial interests.

