## [Peer Review File · Nature Communications]

Reviewers' Comments:

Reviewer #1 (Remarks to the Author)

The authors have added some experimental results that strengthen the study and further support their conclusions. The work improves our understanding of how sequence variants in microRNA genes can alter processing and production of mature miRNAs, potentially affecting normal cell function and altering disease risk. I have only have a few minor issues that could be addressed.

It is misleading to say that the variant is found in breast cancer patients (line 89) as it seems from the reference that is cited for the study reporting the variant, there was only one breast cancer patient found with the variation. The study on gastric cancer had more patients with the A variant so this line could simply be changed to "breast and gastric cancer" for example.

The broader implication of the study for cancer risk could be expanded on in the discussion. The authors could further consider the study by Mu and Su suggesting that the AA allele is associated with poorer outcomes in gastric cancer patients and how their current findings may offer a potential mechanistic explanation.

REVIEWERS' COMMENTS:

Reviewer #1 (Remarks to the Author):

The authors have added some experimental results that strengthen the study and further support their conclusions. The work improves our understanding of how sequence variants in microRNA genes can alter processing and production of mature miRNAs, potentially affecting normal cell function and altering disease risk. I have only have a few minor issues that could be addressed.

It is misleading to say that the variant is found in breast cancer patients (line 89) as it seems from the reference that is cited for the study reporting the variant, there was only one breast cancer patient found with the variation. The study on gastric cancer had more patients with the A variant so this line could simply be changed to "breast and gastric cancer" for example.

The reviewer is indeed correct. We had referred to breast cancer patients, since this was the initial study. We have now corrected the statement, as suggested by the reviewer and have changed the text accordingly

"Here, we investigated the mechanism by which **a** pri-mir-30c-1 variant detected in **breast and gastric cancer patients** results in an increased expression of this miRNA"

The broader implication of the study for cancer risk could be expanded on in the discussion. The authors could further consider the study by Mu and Su suggesting that the AA allele is associated with poorer outcomes in gastric cancer patients and how their current findings may offer a potential mechanistic explanation.

We have partially followed this suggestion. The major emphasis of our study is to show **mechanistically** how a single SNP can drive a change in RNA structure, leading to increased levels of trans-acting factor/s (in this case SRSF3) and promote miRNA biogenesis. This is important for the field, most notably due to the increased interest for a role of SRSF3 in miRNA biogenesis. The role of miR-30c in cancer has not yet been fully uncovered; yet, the finding of a SNP leading to increased expression of mature miR-30c in patients with two types of cancer is highly suggestive. We have expanded this in the Discussion section, as suggested by the Reviewer and are also referring to the poorer outcomes found in patients with the AA allele.